# Anaerobic Power Assessment in Athletes: Are Cycling and Vertical Jump Tests Interchangeable?

**DOI:** 10.3390/sports8050060

**Published:** 2020-05-09

**Authors:** Micah Gross, Fabian Lüthy

**Affiliations:** Swiss Federal Institute of Sport, 2532 Magglingen, Switzerland; fabian.luethy@baspo.admin.ch

**Keywords:** force plate, mechanical power, lower extremity, loaded squat jump, explosive strength, muscle power

## Abstract

Regularly assessing anaerobic power is important for athletes from sports with an explosive strength component. Understanding the differences and overlap between different assessment methods might help coaches or smaller-scale testing facilities maximize financial and temporal resources. Therefore, this study investigated the degree to which cycling sprint and vertical jump tests are interchangeable for determining peak mechanical leg power output in strength-trained athletes. Professional skiers (n = 19) performed unloaded squat jumps (SJ) and other jump forms on a force plate and a six-second cycling sprint (6sCS) test on an ergometer on six occasions over two years. Along with cross-sectional correlations between cycling and jumping power, correlations between longitudinal percent changes and agreement between magnitude-based inferences about individual changes were assessed. Among the tested jump forms, SJ reflected 6sCS best. However, despite extremely large cross-sectional correlation coefficients (0.92) between 6sCS and SJ, and moderate (Pearson’s r = 0.32 for 6sCS with SJ over one-year time spans) to large (r = 0.68 over shorter time spans) correlation coefficients on percent changes, magnitude-based inferences agreed in only around 50% of cases. Thus, for making qualitative assessments about the development of anaerobic power over time in athletes, cycling sprint and squat jump tests are not interchangeable. Rather, we recommend employing the test form that best reflects athletes’ strength and conditioning training.

## 1. Introduction

In various sports—certain track and field events and cycling disciplines, gymnastics, combat sports, and most game and snow sports—explosive actions, such as jumping, accelerating, changing direction, or launching an object or opponent, contribute crucially to performance [1,2,3,4,5,6]. Actions such as these depend heavily on the ability to generate muscle and external force at high velocities and within timespans ranging from several milliseconds to a few seconds [3,7,8,9,10,11]. This ability can be referred to as explosive strength [12]. Since the coupling of force and velocity implied here is well represented by mechanical power, and because the energy for muscle work during these short-duration actions is not generated aerobically, the ability can also be referred to as muscular power [13] or anaerobic power [14]. 

Within the context of performance testing and the monitoring of athletic training, methods must be established for quantifying certain physical abilities considered to be relevant to a particular sports performance [15], to recognize changes in these over time induced by maturation, training, detraining, injury, and the like. In this context, the assessment of anaerobic power has received continuous attention in sports science literature for several decades [16,17,18]. Two of the most prominent methods for assessing anaerobic power that have emerged from past research are various forms of vertical jumping and cycling sprint tests [16,18,19]. The validity and reliability of both test forms are generally accepted [15,20,21,22], and their application extends to a wide range of sports beyond jumping and cycling disciplines alone [23,24,25,26,27,28]. 

Whereas earlier versions of the vertical jump test focused on jump height [16], mechanical power has emerged in recent years as the superior parameter for assessing explosive strength of the lower extremities in athletes [23]. Although jump height is directly coupled to mechanical work performed during the push-off phase and to take-off velocity [29], this parameter does not reflect the time component of muscular force development [30], which is especially important in many sport settings [3,7,8,9,31]. Since mechanical power does consider the time component, it seems fitting that this parameter has shifted to the forefront of consciousness regarding explosive strength assessment. 

This change has come hand in hand with the rapid evolution of testing technology, especially force plates, whose implementation is becoming commonplace in performance testing facilities. High-resolution force plates (sample rates typically ≥1000 Hz) allow the determination of instantaneous mechanical power as the product of instantaneous force and velocity, the latter obtained by mathematical integration of the mass-normalized net force (i.e., acceleration) signal. While force plates capable of measuring in either one or three dimensions now represent the gold standard for determining mechanical power in movements, such as jumping [29], hopping [32], and sprinting [33,34], other technologies have emerged as well. For example, portable position transducers and accelerometers capable of estimating power from acceleration and mass are becoming common for training and field-testing purposes [35,36]. Even a simple, yet remarkably valid, method for determining lower-extremity muscle power based on vertical jump flight time and push-off distance, measured with either a smart phone or other device, has been established [29,37]. 

Although measuring mechanical power in cycling sprint tests has a longer history [17,38], the advent of electronic direct-force cycling power meters capable of recording instantaneous power one or more times per second has refined diagnostics of maximal cycling power output drastically. Further, with the increasingly common use of power meter-equipped bicycles and ergometers in training, athletes and coaches are perhaps beginning to think of cycling sprint performance more in terms of power than ever before [39,40,41].

In addition to developments in measurement technology and criterion parameters, testing procedures themselves have also evolved over the years. Whereas early studies on anaerobic power employed rather long (up to 30 s) cycling sprint tests [17] and quite generic “jump height” tests [16], these have gradually lost favor among researchers and performance testing facilities, and been replaced by shorter sprints [42,43,44] and more standardized jump tests [22]. In particular, the six-second cycling sprint has established itself as a means for determining maximal cycling power in research settings [20,42] and is also commonly employed in standardized performance testing [44]. During all-out cycling sprint tests, peak power is achieved within the first few seconds [17]; thus, longer durations, although appropriate for assessing anaerobic capacity or fatigue resistance, are unnecessary for determining peak power and perhaps even detrimental if they cause subjects to unconsciously pace themselves for the longer effort [21,44]. For assessing jumping power, standardized countermovement or squat jumps without arm movement have become most common. Eliminating arm movement, typically by placing hands on the hips, provides a more reliable measurement of lower extremity power [22]. The squat jump, in particular, is arguably the most elementary and standardizable jump form for assessing anaerobic power of the lower limbs, since it isolates concentric activation over a pre-defined range-of-motion and eliminates the influence of a stretch-shortening cycle [45]. In the case of both cycling sprint and jump tests, improvements have been made over time in the reliability and specificity, and arguably the time-efficiency, of methods, criteria which are essential for assessing and tracking athletes’ performance.

Although both cycling sprint and vertical jump tests are valid and well-established methods for assessing anaerobic power, they obviously differ considerably, particularly in their respective movement patterns (unilateral versus bilateral, and cyclic versus acyclic) and test durations (several seconds versus less than one second). Whereas several previous studies have addressed correlations between anaerobic cycling power and vertical jump height [19,24,25,38,45,46], the aforementioned evolution of methodology and consciousness make power-to-power the more obvious, and thus the more relevant comparison, today than was perhaps the case years ago. However, due to a surprising sparsity of published research on this topic [45,47], it remains unclear how well the power measurements from both tests—and more importantly with regards to performance testing, changes in these—correlate with one another. Information about this relationship is of particular relevance for performance testing centers seeking the most economical use of their spatial and financial resources, as well as athletes’ time and energy. If the two tests are interchangeable, it could be unnecessary for centers to provide facilities for both.

For the two tests to be considered interchangeable, more than a high correlation in cross-sectional data is necessary. It must also be shown that changes in test results over time (longitudinal data), especially those induced by training or de-training, correlate well with one another, and, ultimately, that similar conclusions about training effectiveness would be drawn from both test forms. Thus, the aim of this study was three-fold: The preliminary aim was to re-assess the relationship between anaerobic cycling and jumping performance from a cross-sectional perspective, this time using peak power from both tests, in a group of strength-trained athletes. Building on this, we then investigated, from a longitudinal perspective, the relationship between changes in peak cycling power and in peak jumping power over time. Finally, we assessed how well magnitude-based inferences about individual changes in ability over time agree between the two tests. The corresponding hypotheses to be tested were that cycling and jumping power correlate very strongly, that changes in these correlate very strongly, and that inferences about individual changes in these agree in the large majority (at least 75%) of cases. To fulfil the study aims, we used data from a six-second cycling sprint test and an unloaded vertical squat jump, obtained from professional ski cross racers during their routine performance testing, at six different time points over the course of two annual training cycles. Additionally, because other jump forms were also included in the performance tests at each time point, we explored analogous relationships between cycling peak power and both countermovement jump power and squat jump peak power with an additional load equal to body weight.

## 2. Materials and Methods

### 2.1. Subjects

Data were collected from professional ski cross racers from the Swiss national team, over the course of two annual training cycles, during regular performance testing at the Swiss Federal Institute of Sport (Magglingen, Switzerland). To be precise, tests were conducted in the months of May, August, and November of two consecutive years. Estimates of subjects’ one-repetition maximum for back half squats (knee angle ~100°) based on isometric squatting against a force plate and a validated conversion factor of 0.7 [48] were 2.0–2.7 × body mass for females and 2.2–3.4 × body mass for males. Thus, subjects were considered to be strength-trained. Although the primary reason for the performance testing was to guide the athletes individually in their training, all athletes gave written consent for their anonymized data to be used for research purposes. Further, the methods and procedures of the research were approved by the ethical review board of the canton of Bern, Switzerland (project ID 2018-00742). In total, data from 20 athletes were available for the study. Descriptive characteristics of the cohort at the onset of the study are displayed in Table 1.

### 2.2. Data Collection

Because data were obtained during routine performance testing, additional measurements not included in the present investigation were also performed. Upon arriving at the testing center, athletes’ body height was measured using a stadiometer. Athletes then proceeded to warm up individually for 20–30 min. Testing commenced with an isometric squat strength testing procedure (data not included in the study), which lasted around five minutes. Thereafter, athletes completed a battery of single vertical jumps, including countermovement and squat jumps with additional loads ranging from 0% to 100% of body weight. For the unloaded condition, athletes placed their hands at the hips to eliminate an arm swing while jumping. For loaded conditions, athletes jumped with a 10-kg barbell loaded with weight plates placed across their shoulders (as when performing back squats). A custom-made, hand-activated retention system, which was engaged during the flight phase of loaded jumps, ensured that subject could safely land with no load on their shoulders. Squat jumps were commenced from a static starting position with a knee angle of approximately 90°, which was controlled visually by an investigator. For countermovement jumps, depth of the countermovement was determined instinctively by the athletes themselves (typically 0.20–0.35 m), but data were filtered before analysis (see below) to ensure continuity in individuals’ jump execution across time points. Athletes were instructed to jump as high and explosively as possible. For unloaded jumps and jumps with additional loads up to 20% of body weight, three valid trials were performed. For heavier loads, to minimize fatigue, only one valid trial was required, although 1–2 additional trials were performed in the case of obviously poor execution or peak power values clearly lower than expected based on the trend of preceding loading condition. 

Jumps were performed on a one-dimensional force plate (MLD Test EVO 2, SP Sport, Trins, Austria), which recorded ground reaction force at 1000 Hz. Using the total mass, determined by the force plate prior to each jump, the accompanying software (Muskelleistungsdiagnose 2010, version 5.2.0.6101, InfPro IT Solutions GmbH, Innsbruck, Austria) calculated acceleration–time, velocity–time, power–time, and position–time curves of the center of mass from the recorded force–time signal for each jump. Squat jumps for which mechanical power descended below −1 W/kg body mass within 1 s prior to the onset of positive (upward) velocity—indicating an unacceptable countermovement—were deemed invalid and deleted. 

For the present investigation, we decided a priori to include only squat jumps (because these are inherently most standardized) performed with no additional load (because this is most common in practice), to keep the results concise. However, during data analysis, the other extreme loading condition (100% body weight) as well as countermovement jumps (unloaded only) were deemed helpful for explaining findings, and were thus included in the presentation of results. Thus, the jump parameters retained for data analysis were the peak concentric power for squat jump (SJ), squat jump with an additional load equal to body weight (LSJ), and unloaded countermovement jump (CMJ). For each jump type, the value from the trial with the highest peak concentric power was retained. The test-retest percent typical error (CV) for peak power using the described protocol has been previously determined to lie between 2.7% (unloaded) and 4.7% (100% body weight additional load) for squat jumps and between 2.5% (unloaded) and 3.9% (100% load) for countermovement jumps [49]. 

Following the last vertical jump, athletes proceeded to the six-second cycling sprint test. This was performed with a flying start on a Wattbike ergometer (Wattbike Trainer, Nottingham, UK) according to the 6” Peak Power Test protocol in the Wattbike test guide [50] (pp. 19–21). Initially, saddle height and handlebar position were set up according to athletes’ personal preferences (settings were determined at the first time point and replicated thereafter). Then, athletes pedaled with minimal resistance (<100 W) for 1–2 min while the test procedure was explained to them by an investigator. Resistance settings for the test were determined at each time point anew based on sex and current body mass, according to the recommendations in the Wattbike test guide [50] (p. 24). Precisely, according to body mass ranges for each gender (see Table 1), the air/magnet settings varied between 4/1 and 6/1 for females and between 8/1 and 8/6 for males. Immediately prior to the measurement, athletes pedaled with no load for 20–30 s while the protocol was selected on the Wattbike monitor. Simultaneously with the onset of the six-second timer, the resistance was increased to the selected level by an investigator, and athletes pedaled as hard as fast as possible for six seconds while seated on the saddle. During the test, the Wattbike was stabilized by an investigator or coach, to prevent unwanted movement of the entire ergometer. The Wattbike measures chain tension with a load cell at 100 Hz and crank angular velocity twice per revolution yielding two power values per revolution [51,52]. For the present study, only the highest power value recorded during the six-second cycling sprint test (6sCS) was analyzed.

### 2.3. Statistical Analysis

Statistical analyses were performed using customized Python scripts. Initially, to ensure continuity of individuals’ jump execution across time points, jumps for which the concentric push-off distance differed by more than 0.05 m from the subject’s mean value were excluded from data analysis. For descriptive purposes, peak power values averaged across all time points for each of the four tests (6sCS, SJ, LSJ, CMJ) were assessed for normal distribution using the Shapiro–Wilk test and a significance level (α) of 0.05, then compared for females, males, and the pooled cohort using either a one-way ANOVA (normally distributed data) or the Kruskal–Wallis test (if one or more tests did not pass the test for normal distribution), also with α = 0.05. Post-hoc comparisons between tests were made using Bonferroni correction. 

For hypothesis testing, change scores were calculated and expressed as a percent of the pre-test value, rather than in Watts. With six time points, a maximum of 15 ($\sum_{t=1}^{5}t$) possible change scores per subject could be calculated. For additional analyses, two particular sub-types of changes were analyzed separately: changes between consecutive time points (t_2_-t_1_, t_3_-t_2_, t_4_-t_3_, t_5_-t_4_, t_6_-t_5_, five total) and changes between time points separated by one year (t_4_-t_1_, t_5_-t_2_, t_6_-t_3_, three total), because these represent two common time spans between performance tests for elite and amateur athletes. Peak power and change score data sets were tested for normal distribution using the Shapiro–Wilk test (α = 0.05). Thereafter, relationships between raw power values (cross-sectional correlations) and between change scores (longitudinal correlations) were assessed using either Pearson’s r (if both data sets were normally distributed) or Spearman’s rho (if one or both data sets did not pass the test for normal distribution). For cross-sectional correlations, both absolute and body-mass-normalized power were analyzed, since absolute power was expected to be rather heterogeneous, which can exaggerate the strength of correlations. For the same reason, cross-sectional correlations were performed separately for males and females. Longitudinal correlations, however, were performed on change scores for absolute power only and for both genders combined; this choice was made for the sake of being concise while maximizing *n*, and since the effects on percent changes of gender and normalizing power were expected to be negligible. Correlation coefficients were categorized based on their magnitude as trivial (0–0.1), small (0.1–0.3), moderate (0.3–0.5), large (0.5–0.7), very large (0.7–0.9), or extremely large (>0.9) according to Hopkins et al. [53]. 

Finally, a magnitude-based inference about each individual change was made using the methods described by Hopkins [54] and integrated into his open-source spreadsheet [55]. For SJ, LSJ, and CMJ peak power, percent typical errors (CV) of 2.7%, 4.7%, and 2.5%, respectively, were used [49], whereas a 4.9% typical error was assumed for 6sCS peak power [52]. For all inferences, the smallest meaningful change was set to 1%, and the confidence range for the true change was 80%. In short, based on measured changes, typical errors, and smallest meaningful changes, inferences were formulated, each of which included a degree of certainty (‘very likely’, ‘possible’, or ‘unclear’) and a change type (‘increase’, ‘trivial change’, or ‘decrease’). To assess the degree of agreement between magnitude-based inferences, a three-tier approach was used: The percentage of cases was reported for which (1) the most likely change (independent of its certainty) agreed, (2) the phrased inferences agreed, and (3) phrased inferences contradicted. Inferences were considered to agree if the most likely changes agreed and certainty of both were not ‘unclear’. Inferences were considered to contradict if one included ‘increase’ and the other included ‘decrease’ and certainty of both were not ‘unclear’. 

## 3. Results

### 3.1. Dataset Description

In all, data from 19 athletes (six females, 13 males) yielded relevant change scores and were thus included in data analysis. Fifty-two test sessions from 16 different athletes (6 females, 10 males) included all four peak power measurements, thus allowing direct comparison between tests. These 52 sessions are summarized in Figure 1 and Table 2. As seen in Figure 1, 6sCS differed significantly from all jump tests, and LSJ differed significantly from both SJ and CMJ (same for females, males, and pooled cohort).

Correlations with 6sCS peak power values were possible using a total of 77 coincident data points (13 ± 3 per time point) for SJ and CMJ and 69 (12 ± 3) coincident data points for LSJ. The total number of jumping power change scores with a coincident 6sCS change score was 136 for SJ and CMJ and 107 for LSJ. The subsets of change scores between two consecutive time points contained 49 values (10 ± 3 per inter-test period) for SJ and CMJ and 44 (9 ± 3) for LSJ. The subsets of change scores between time points separated by one year comprised 24 values (8 ± 3 per inter-test period) for SJ and CMJ and 21 (7 ± 3) for LSJ. 

### 3.2. Summary of Correlations

#### 3.2.1. Power Values (Cross-Sectional Data)

Pooling all six time points, correlation coefficients between absolute cycling peak power and absolute jumping peak power were very large or extremely large for females (0.72–0.81) and males (0.78–0.90). However, correlation coefficients using body-mass-normalized peak power values were moderate to large for females (0.38–0.58) and large to very large for males (0.52–0.71). These and additional correlation data are displayed in Table 3 and Figure 2 and Figure 4.

#### 3.2.2. Change Scores (Longitudinal Data)

Correlation coefficients between coincident SJ and 6sCS change scores from all possible time point combinations or consecutive time points were large. For time points one year apart, change scores correlated only moderately. Corresponding values for CMJ were similar, albeit lower, whereas LSJ correlation coefficients depended less on time span. These and additional correlation data are displayed in Table 4 and Figure 3 and Figure 4.

#### 3.2.3. Magnitude-Based Inferences

When probabilities of different types of changes between two consecutive time points were calculated for 6sCS and SJ, the most likely type of change agreed in the majority (63%) of cases; for changes across one year, the most likely change type agreed in slightly less than half (48%) of cases. Actual inferences agreed in around half of all cases (43%–54%) and contradicted often (17%–26%), only slightly dependent on time span. The corresponding percentages based on all time point combinations were within the same ranges as for the aforementioned subsets. These results, as well as those from other test pairs, including LSJ and CMJ, are displayed in Figure 5.

## 4. Discussion

This study aimed to investigated the degree to which a six-second cycling sprint test performed on a Wattbike ergometer and a vertical squat jump test on a force plate are interchangeable as means for assessing anaerobic power of the lower extremities in strength-trained athletes. Answering this question could provide helpful information for smaller performance testing facilities, as well as coaches or sports clubs, who intend to invest in testing devices and wish to do so in the most resourceful manner possible. The main findings were that (1) extremely large linear correlations exist between maximal cycling power and maximal jumping power (Table 3, Figure 2 and Figure 4), (2) large linear correlations on individual percent changes in maximal power across time spans of less than one year exist between cycling and unloaded jumping (Table 4, Figure 3 and Figure 4), and (3) magnitude based inferences made about individual test-to-test changes from sprint cycling and squat jumping power agree in around half of all cases, whereas the most-likely types of change (substantial increase, decrease, or trivial change) agree in a slight majority of cases (Figure 5).

First, this study confirmed a strong relationship between jumping power and sprint cycling power in the current cohort of strength-trained ski cross racers. Whereas previous studies have shown similar relationships, usually using jump height rather than power, in untrained subjects [46], strength-trained non-cyclists [19,24,25,38,56], and strength-trained cyclist [27], this is one of few studies [45,47] to have directly compared mechanical power from both test forms. Thus, the close relationship between the two types of power measures seems to be fairly universal. Errors in the estimate of squat jumping maximal power from sprint cycling power using the regression equation from Figure 2a were 434 (90% c.l. 383–502) W or about 10%. Similarly, strong relationships with 6sCS were found for LSJ and CMJ. These relationships are not surprising and, as has been described elsewhere, can be attributed to the fact that power in both tests is mainly produced by the same muscle groups (hip and knee extensors) in a similarly dynamic fashion. The clear systematic differences in the actual power values (Table 2, Figure 1) are most obviously due to the fact that cycling is a unilateral movement, whereas jump tests were performed in a bilateral manner. However, the magnitude of the differences between cycling and all forms of jumping suggests that other factors are in play as well. One such factor could have been diminished coordination for the cyclic movement [57]. Another explanation for the lower power values in the 6sCS (<500 W in a few cases) was the combination of low ergometer resistance and a flying start, an effect that was exacerbated in the case of athletes with particularly steep force–velocity profiles (i.e., high relative maximal strength but comparatively low maximal contraction velocity). In essence, the movement velocity was seemingly too high for some athletes, and peak power was limited accordingly. Indeed, it pilot measures performed after the study with some of the same athletes, cycle sprint tests with higher resistance and/or a standing start yielded 5%–10% higher peak power values. This poses the question of whether the test protocol and resistance settings recommended by Wattbike should be applied to strength-trained athletes, which might merit further exploration in practical settings.

For sprint cycling and vertical jump tests to be considered interchangeable, a predictable relationship must exist not only for cross-sectional data but also between changes in performance over time. To our knowledge, this relationship has not been well established in previously published research; doing so was, therefore, an important aim of this study. Correlations between longitudinal percent changes in power were somewhat smaller than the cross-sectional correlations on actual power, but nonetheless moderate to large in magnitude. As expected when designing the study, correlations between jumping and cycling power change scores were highest for SJ; nonetheless, 6sCS change scores also yielded large correlation coefficients when paired with LSJ or CMJ change scores. In general, the weakest correlations between jumping and cycling change scores were found for time spans of one year. One reason for the smaller correlations in this subset of change scores was the smaller range of values, which is not surprising since athletes were in the same phase of physical preparation at both time points. This finding suggests that larger changes in one test may be reflected in the other test, whereas smaller changes are not. 

As a point of comparison, changes in power from the various jumping forms (SJ, LSJ, and CMJ) between consecutive time points also correlated slightly better (0.73–0.76) than those occurring over one year (0.59–0.71). Further, although change scores from the various jump forms correlated more strongly with each other than did change scores from any jump form with those from cycling, the differences were not all too drastic. Thus, from a longitudinal perspective, it appears that sprint cycling and squat jumping power tests reflect one another only about as well as do squat jump and countermovement jump tests. Typically, however, squat and countermovement jump tests are not considered interchangeable.

Although observed changes were mostly within around ±9%, it might seem remarkable to some readers that professional athletes occasionally experienced changes of almost ±20% within a few months. However, we do not find such changes particularly surprising considering that the present subjects’ (skiers’) explosive strength is generally only moderate (compared to sprint specialists, for example). At the same time, the present subjects (and skiers in general) possess very good maximal strength, which affords their explosive strength a rather large degree of trainability during the off-season conditioning phase. Additionally, a competitive season and a post-season break lay between time points 3 and 4, which could explain most of the larger decreases in power we observed. The occasional extreme changes, especially between consecutive time points, certainly expanded the data range, which tends to increase correlation coefficients. Accordingly, we attribute the lower correlation coefficients between change scores over one year to the narrower range of change scores (due to the fact that athletes were at the same phase of their physical preparation period at both time points).

One of the main uses of performance testing is for tracking changes in an athlete’s physical abilities over time. In the most rudimentary sense, this means breaking down test results from two time points into one of three possible concrete conclusions: the tested ability improved, it declined, or no meaningful change occurred. Magnitude-based inferences on individual changes are a statistical method that uses known or assumed values for measurement precision and the smallest meaningful change in a given ability to infer (with a precluding degree of certainty) one of these three conclusions. If such inferences are to be made about an individual’s anaerobic power based on either SJ or 6sCS, and if the two tests are to be considered interchangeable, the probability that the inference over the same time span from the other test would agree must be high. Moreover, there must be a very small probability that inferences from the two tests would contradict. In the present study, these criteria were not fulfilled to a satisfactory degree. In the case of changes in 6sCS compared to changes in unloaded SJ, the most likely change (i.e., substantial increase, substantial decrease, or trivial change) displayed the most frequent agreement, in up to 63% of cases. However, since the most likely change does not always differentiate itself from the other two possible changes with enough certainty to justify its ultimate adoption, worded inferences agreed in only 43%–54% of cases, somewhat dependent on the time span. Moreover, inferences contradicted in 17–26 cases. For other jump forms, agreement with 6sCS was more or less the same. To put this result into context, around 20% agreement could be expected with random guessing alone. For comparison, inferences for SJ and LSJ agreed more frequently (57%–67% of cases) and also contradicted less often (11%–13% of cases). We must concede, however, that the agreement in the large majority of cases we were looking for (75%, see hypotheses) was not achieved among the various jump tests either. Interestingly, inferences for SJ and CMJ agreed the least frequently (~33% of cases) among the tests we analyzed. Therefore, based on magnitude-based inferences, the degree of interchangeability between squat jump and cycle sprint tests is not particularly high, but comparable to the degree of interchangeability between various jump execution forms.

We recognize that the percentages of agreeing and contradicting inferences attained with our methods are dependent on choices we made regarding confidence intervals and smallest important changes; however, based on simulations with our data, percentages would not have been uniformly improved with another confidence range between 70% and 90%. Further, increasing the smallest important percent change from 1% up to 5% would have decreased percentages of both agreeing and contradicting cases simultaneously. Therefore, we believe our conclusions would not have been different had other reasonable choices for confidence intervals and smallest important changes been made.

## 5. Conclusions

This study confirmed very large correlations between various forms of jumping power and sprint cycling power in strength-trained professional ski cross racers. Changes in the two power assessments over time correlated less strongly, although still rather well, especially for unloaded squat jumps and time spans of less than one year. Nonetheless, where qualitative inferences were made based on performance changes in one test, there was a level of agreement with the other test, which we consider less than acceptable. In the case of the present subjects, whose strength and conditioning training was more closely replicated by the squat jump tests, we would be skeptical about drawing conclusions about their conditional development based on the cycling sprint measures alone. Therefore, we do not recommend using the cycling sprint and squat jump tests interchangeably for monitoring anaerobic power of the lower extremity, but rather to opt for the test form that most closely replicates athletes’ strength and conditioning training.

## Figures and Tables

**Figure 1 sports-08-00060-f001:**
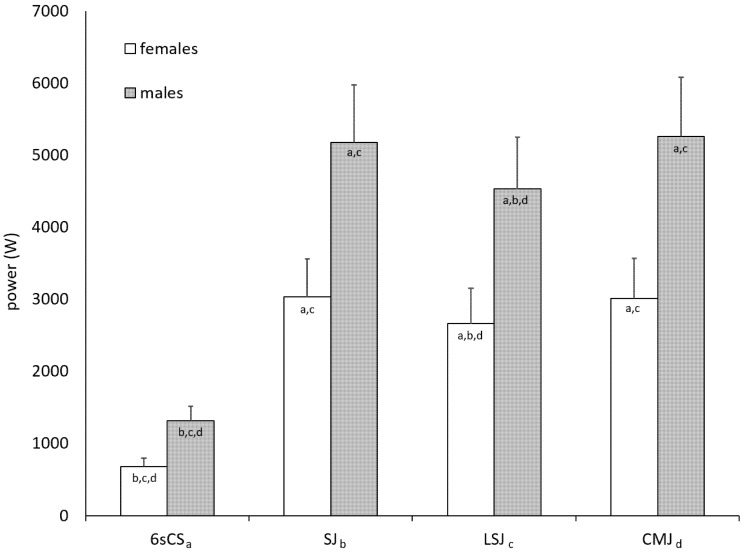
Group mean and standard deviation for peak power by test form. Data taken from 52 complete test session, i.e., those in which a subject performed all four tests. 6sCS: six-second cycling sprint. SJ: unloaded squat jump. LSJ: squat jump with additional load equal to body weight. CMJ: unloaded countermovement jump. Letters in the columns designate a significant difference with the test bearing that subscript on the horizontal axis.

**Figure 2 sports-08-00060-f002:**
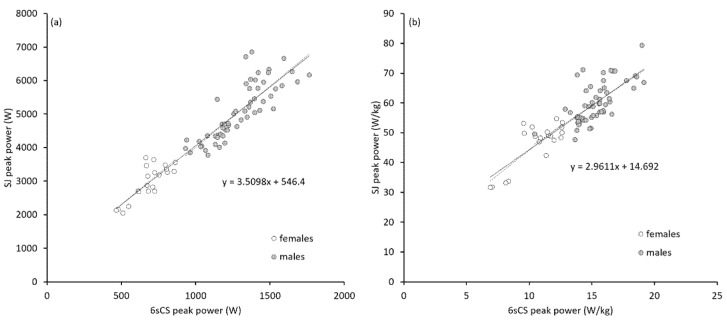
Linear relationships and regression equations between absolute (**a**) and body-mass-normalized (**b**) peak power from the six-second cycling sprint (6sCS) and unloaded squat jump (SJ) tests. Dashed regression lines are for female and male subgroups. Solid regression line and displayed regression equations are for males and females pooled together.

**Figure 3 sports-08-00060-f003:**
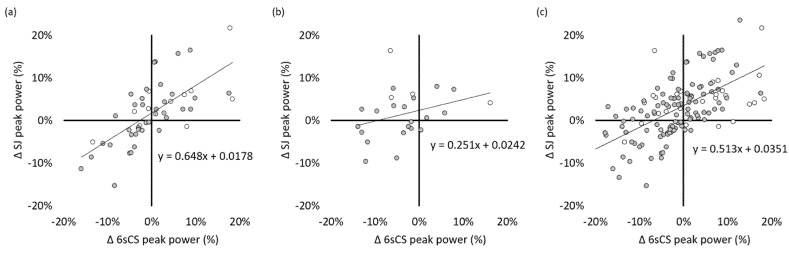
Linear relationships with regression equations between percent changes (Δ) in peak power for the six-second cycling sprint (6sCS) and unloaded squat jump (SJ) tests; (**a**) changes between consecutive time points; (**b**) changes between time point one year apart. (**c**) changes between all time point combinations. Empty circles represent females. Filled circles represent males. Regression lines are for females and males pooled together.

**Figure 4 sports-08-00060-f004:**
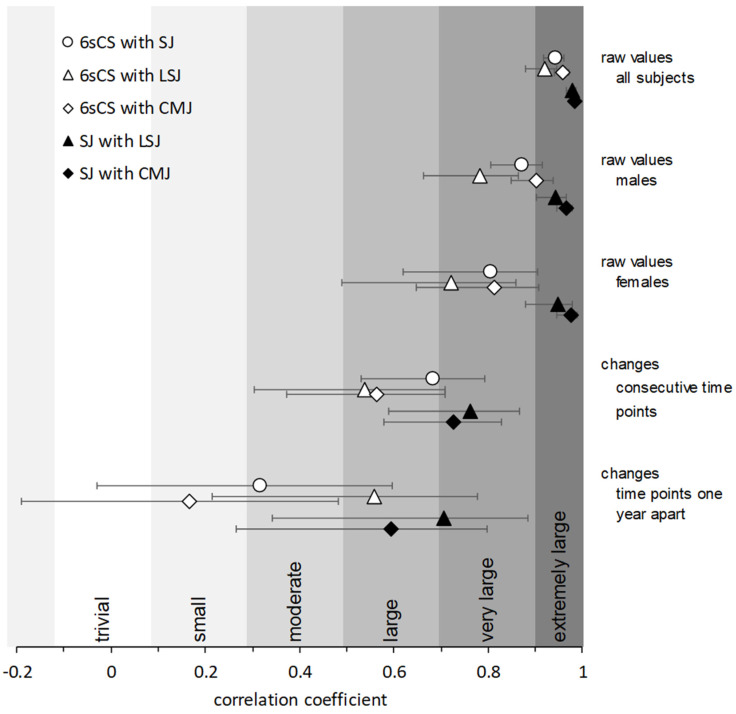
Correlations coefficients and classifications for absolute peak power values (by gender) and percent change scores in peak power by time span. Error bars represent 90% confidence intervals. 6sCS: six-second cycling sprint. SJ: unloaded squat jump. LSJ: squat jump with additional load equal to body weight. CMJ: unloaded countermovement jump.

**Figure 5 sports-08-00060-f005:**
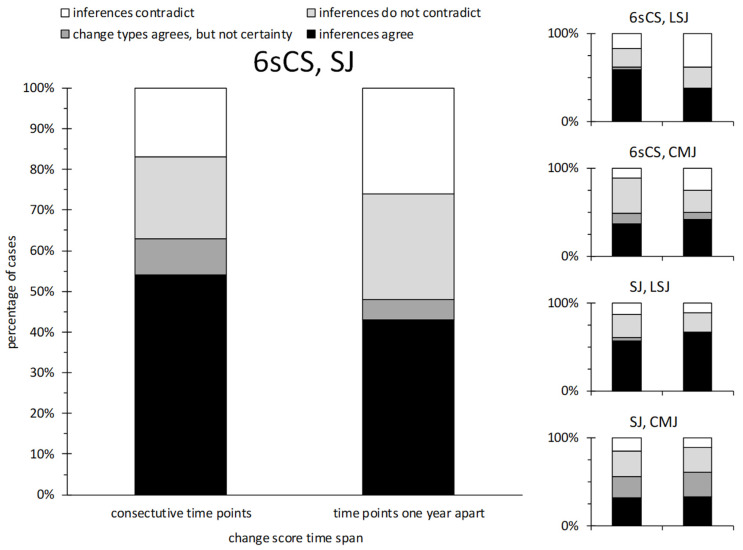
Agreement of most probable changes and magnitude-based inferences about individual changes in peak power between cycling sprint and vertical jump tests. Plot titles indicate the two compared tests. Axes and colors of the smaller plots (comparisons of secondary importance) correspond to those of the main plot. 6sCS: six-second cycling sprint. SJ: unloaded squat jump. LSJ: squat jump with additional load equal to body weight. CMJ: unloaded countermovement jump.

**Table 1 sports-08-00060-t001:** Descriptive characteristics of the subject cohort at the onset of the study.

Group	n	Age (y)	Body Height (m)	Body Mass (kg)
females	7	23 ± 5 (18–31)	1.68 ± 0.04 (1.63–1.74)	63 ± 5 (56–71)
males	13	25 ± 4 (18–31)	1.83 ± 0.09 (1.71–2.01)	87 ± 10 (70–110)
all	20	25 ± 4 (18–31)	1.78 ± 0.1 (1.63–2.01)	79 ± 14 (56–110)

Data are presented as mean ± standard deviation (range).

**Table 2 sports-08-00060-t002:** Anaerobic power descriptive data.

Group		6sCS		SJ		LSJ		CMJ	
	N *	W	W/kg **	W	W/kg **	W	W/kg **	W	W/kg **
females	6	680 ± 118	10.5 ± 1.9	3030 ± 528	47 ± 8	2660 ± 493	41 ± 8	3014 ± 554	47 ± 7
males	10	1317 ± 197	15.5 ± 1.5	5179 ± 797	61 ± 6	4528 ± 720	53 ± 6	5262 ± 820	62 ± 6
all	16	1121 ± 345	14.0 ± 2.8	4518 ± 1233	57 ± 94	3953 ± 1089	50 ± 9	4570 ± 1284	57 ± 10

Data are peak power values from the different tests (averaged over all time points) and are presented as mean ± standard deviation. * Data taken only from 52 sessions in which subjects performed all tests. ** Power normalized to body mass. 6sCS: six-second cycling sprint. SJ: squat jump. LSJ: squat jump with an additional load equal to body weight. CMJ: unloaded countermovement jump.

**Table 3 sports-08-00060-t003:** Cross-sectional correlation coefficients between anaerobic power values from different tests.

Group	6sCS, SJ		6sCS, LSJ		6sCS, CMJ		SJ, LSJ		SJ, CMJ	
	W	W/kg *	W	W/kg *	W	W/kg *	W	W/kg *	W	W/kg *
females	**0.81**	***0.58***	**0.72**	*0.38*	**0.81**	*0.51*	**0.95**	***0.83***	**0.98**	***0.94***
males	***0.87***	***0.67***	**0.78**	***0.52***	***0.90***	***0.71***	***0.94***	***0.84***	***0.97***	***0.90***
all	**0.94**	***0.84***	**0.92**	***0.76***	***0.96***	***0.88***	**0.98**	***0.90***	***0.98***	***0.95***

Bold font indicates statistically significant correlation. Italic font indicates Spearman’s rho (otherwise Pearson’s r) because one or both data sets did not pass the Shapiro–Wilk test for normal distribution. * Based on power normalized to body mass. Column-pair headings indicate the two tests correlated. 6sCS: six-second cycling sprint. SJ: squat jump. LSJ: squat jump with an additional load equal to body weight. CMJ: unloaded countermovement jump.

**Table 4 sports-08-00060-t004:** Longitudinal correlation coefficients between percent change scores from different tests.

	6sCS, SJ	6sCS, LSJ	6sCS, CMJ	SJ, LSJ	SJ, CMJ
	% Change *	% Change*	% Change *	% Change *	% Change *
consecutive time points	**0.68**	**0.54**	**0.56**	**0.76**	**0.73**
time points one year apart	0.32	**0.56**	0.17	**0.71**	**0.59**
all time point combinations	***0.61***	**0.47**	**0.48**	***0.69***	***0.71***

* percent change scores were calculated from absolute (not body-mass-normalized) power values. Bold font indicates statistically significant correlation. Italic font indicates Spearman’s rho (otherwise Pearson’s r) because one or both data sets did not pass the Shapiro–Wilk test for normal distribution. Column-pair headings indicate the two correlated tests. 6sCS: six-second cycling sprint. SJ: squat jump. LSJ: squat jump with an additional load equal to body weight. CMJ: unloaded countermovement jump.

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
