# Peer review of "Anaerobic Power Assessment in Athletes: Are Cycling and Vertical Jump Tests Interchangeable?"

_sports, 2020, doi:10.3390/sports8050060_

Round 1
Reviewer 1 Report
A clear introduction with a question of practical importance, aimed at reducing the number of different tests that are used to measure anaerobic peak power production.
A strength of the present study is that it seems that the tested athletes were very proficient. However, this needs further corroboration, since the large change scores in quite a few athletes seem odd in this context (see below)
Line 142: 1-3 trials: why was sometimes only one attempt done? How often did this occur, may this have contributed to the sometimes large individual changes scores?
Line 150 slight countermovement. ? Define slight: please mention the vertical counter distance of the CoM in cm that was still valid and included in the analysis. Judgement by eye is fine in practice, but it needs additional verification during data analysis. It is very difficult to execute a SJ without any counter-movement, either in the knee and/or hip joint.
L152 Please explain why these two jump types (and not the others) were selected.
Would the study outcome have been different for other jump-types (I assume that those also have been analysed) please comment on this in the discussion?
Similarly: why were both SJ and LSJ included. It can be foreseen that both jumps are highly correlated. Albeit that reproducibility of a SJ with bodyweight is less. Good execution of the latter is not feasible for many (less proficient) athletes.
160 please provide the resistance values that were used in the text (instead of referring to the manual). I assume that these did not change over time? Or was the resistance adjusted in case of changes in body mass?
164 I assume that the data were filtered provide details about this (also sample frequency etc), what exactly is the highest value? is that the highest data point in the filtered signal (or an average value of a few rotations (pedal strikes)?
167why? Normalization to bodyweight often has been done, this will decrease correlation coefficients.
The reason for using non-normalized data should be provided and the effects of normalization of r-values should be included.
In relation to this: What is the effect of differences in body composition: including male and female athletes also increases between-subjects variability (and thereby increase correlation coefficients)
169 …compared using a one-way ANOVA and paired t-tests. I would have thought that a repeated-measures ANOVA with two within factors (time; 6 levels) and test-type (3 levels) with Bonferroni adjusted postdoc test would be the appropriate test.
181 a percent typical error of 2.5% was used (based on unpublished reliability assessments from our 181 lab), 2.5% doesn’t seem unreasonable. However, it does not seem in line with the often large change scores (Figure 3)
200 please report separate power values for men and woman. It is hard to see, but I am not particularly impressed by the Watt bike peak power values. Please provide more details about the athletes how proficient were they? Were there any junior athletes (who in general may show relatively large progress over the years etc.)
Line 202: this should be a repeated-measures ANOVA with Bonferroni corrected pair comparisons I think (not that the outcome probably will be different)
Figures: please increase Font sizes along the axis for readability
Figure 2 please use clearly different symbols for the data of the men and women, this makes the increase in the between-subject variability that is created by the inclusion of both sexes, visible (which is an honest thing to do) and again 400 Watt peak power for a top athlete, doesn’t seem impressive (even for a woman, but this is hard to judge without normalization)
Please check numbering of tables and figures in the text.
Figure 3 I see changes in performance up to 20%
these are high-levels athletes? 20% changes (or even 10 %) are enormous. even within a few months. Are these athletes recovering from injury or suffering from an injury or extensive periods without training (-20%!)
the typical errors for these jump test only is 2.5% thus these changes seem real, but I never encountered a healthy well-trained athlete who trained for several years to increase jump performance by 10-20 %. Has this, for example, to do with the fact that peak power (instead of average power) was calculated: I can imagine that variability in peak power is larger, because it may be more sensitive to smaller changes in jump-execution (compared to average power and jump height)
Especially since you emphasise in the discussion (line 308) ;
It is worth mentioning here that the present subjects were highly-trained professional athletes, the likes of which experience smaller changes in performance for a given time span than untrained subjects.
Magnitude based inferences depend on choices (typical errors confidence intervals) that one makes. The outcome of the study, that change scores for peak power during cycling and jumping can be different, and the deviation may increase with longer intervals between measurements is clear (and seems not very surprising, but still relevant to investigate). However, the outcome very much depends on if the test- circumstances over the years (between test moments) did not change. These factors (for instance the coach being present or not, a different coach and/or test pilot being present) affect performance. How valid is the assumption of a constant typical error? Could you comment upon the stability of the testing circumstance in the discussion? Are you positive that every individual really performed maximally on every occasion (also because sometimes only 1 attempt seems to have been made)
Most importantly: please include the analysis for the comparison between SJ and LSJ.
Now you compare each of them with the cycling test and in the conclusion, you state that the choice of the test should be made in relation to the athletes’ strength and condition training. (which in the present athletes would be more a jump-type of training).
If you could show that the similarity between changes scores is greater between the two jump-types, compared to between each of the jumps separately compared to cycling, this would seem to strengthen your suggestion.
On the other hand, I wouldn’t be surprised if the similarity of the change scores between both jump types would not be greater at all compared to those in relation to cycling. There always is random variability in these tests that quite easily may become larger (with all kinds of things happening, injury, illness, broken relationships ) than the real changes in performance that can be attained, especially in highly trained athletes in which progress is difficult to achieve.
Author Response
General notes to the revision:
Due to filtering, data vary slightly from those in the original manuscript. Also the criterion for contradicting inferences was re-defined and slightly stricter than before (before only ‘very likely increase’ and ‘very likely decrease’ considered contradicting, whereas now, any ‘decrease,’ regardless of whether inferred to be ‘very likely’ or ‘possible,’ is considered to contradict another inference which includes ‘increase,’ also independent of the certainty qualifier). This change led to slightly different percentages of agreeing and contradicting cases than before. However, the main conclusions of the study remain the same. Also, in response to certain reviewers’ comments, CMJ, which were also measured, are included in the study.
Also, correlations for individual time points and time spans were excluded for conciseness, and because these did not contribute substantially to the study. For the same reason, the original Table 2 was removed (these data were and are still visible in Figure 3).
Reviewer 1
A clear introduction with a question of practical importance, aimed at reducing the number of different tests that are used to measure anaerobic peak power production.
A strength of the present study is that it seems that the tested athletes were very proficient. However, this needs further corroboration, since the large change scores in quite a few athletes seem odd in this context (see below)
Line 142: 1-3 trials: why was sometimes only one attempt done? How often did this occur, may this have contributed to the sometimes large individual changes scores?
Thanks for this comment; this was actually poorly described in the original manuscript. The text now includes the following (new line 156):
“For unloaded jumps and jumps with additional loads up to 20% of body weight, three valid trials were performed. For heavier loads, to minimize fatigue, only one valid trial was required, although 1 – 2 additional trials were performed in the case of obviously poor execution or peak power values clearly lower than expected based on the trend of preceding loading condition.”
And the following (new line 175)
“The test-retest percent typical error (CV) for peak power using the described protocol has been previously determined to lie between 2.7% (unloaded) and 4.7% (100% body weight additional load) for squat jumps and between 2.5% (unloaded) and 3.9% (100% load) for countermovement jumps (Hübner, 2009).”
As for outliers being due to only one trial being performed (LSJ only), we don’t believe this played a significant role. Drastic outliers are usually very apparent during testing and in such cases, further trials were performed.
Line 150 slight countermovement. ? Define slight: please mention the vertical counter distance of the CoM in cm that was still valid and included in the analysis. Judgement by eye is fine in practice, but it needs additional verification during data analysis. It is very difficult to execute a SJ without any counter-movement, either in the knee and/or hip joint.
The software’s automated criterion for valid squat jumps is now specified more precisely in the text (new line 165):
“…in order to ensure continuity of individuals’ jump execution across time points, jumps for which the concentric push-off distance differed by more than 0.05 m from the subject’s mean value were excluded from data analysis.”
Squat jump depth was controlled visually during testing (as previously stated) and verified post-hoc (new in the revised manuscript) based on the concentric push-off distance provided by the force plate software (new line 197):
“…squat jumps with a concentric push-off distance differing by more than 5 cm from the athlete’s mean value were filtered out of the final data set…”.
L152 Please explain why these two jump types (and not the others) were selected.
In response to your and other comments, CMJ are now included to help explain and substantiate our findings. Moreover, the text now includes the following sentence to make the reasoning behind our choice more evident (new line 168):
“For the present investigation, we decided a priori to include only squat jumps (because these are inherently most standardized) performed with no additional load (because this is most common in practice), to keep the results concise. However, during data analysis, the other extreme loading condition (100% body weight) as well as countermovement jumps (unloaded only) were deemed helpful for explaining findings, are were thus included in the presentation of results.”
Would the study outcome have been different for other jump-types (I assume that those also have been analysed) please comment on this in the discussion?
Cross-sectional correlations when countermovement jumps (unloaded and loaded) were substituted for squat jumps were very similar, and produced the same conclusions. Correlations on change scores were always lower for CMJ than for SJ, sometimes substantially, sometimes within the same classification. In any case, in response to your comment and those of others, CMJ are now included along with LSJ as secondary parameters. As you mention in a comment below, comparing SJ to LSJ and now CMJ provide some sort of comparison about how much agreement could be expected between two quite similar tests.
Similarly: why were both SJ and LSJ included. It can be foreseen that both jumps are highly correlated. Albeit that reproducibility of a SJ with bodyweight is less. Good execution of the latter is not feasible for many (less proficient) athletes.
The inclusion of LSJ was indeed not necessary to address the study’s primary aim, and we agree that your question is justified. It should be clearer now that LSJ and (now also) CMJ are of secondary importance and serve to substantiate findings.
160 please provide the resistance values that were used in the text (instead of referring to the manual). I assume that these did not change over time? Or was the resistance adjusted in case of changes in body mass?
The resistance was determined anew at each time point, as would be the case in practice (note: with changes in body mass, the resistance for unloaded and loaded jumps changes as well). The range of resistance settings is now stated explicitly in the text as follows (new line 184):
“Resistance settings for the test were determined at each time point anew based on sex and current body mass, according to the recommendations in the Wattbike test guide [50, p. 24]. Precisely, according to body mass ranges for each gender (see Table 1), the air/magnet settings varied between 4/1 and 6/1 for females and between 8/1 and 8/6 for males.”
164 I assume that the data were filtered provide details about this (also sample frequency etc), what exactly is the highest value? is that the highest data point in the filtered signal (or an average value of a few rotations (pedal strikes)?
Essentially, the Wattbike records power twice per crank revolution and we used the highest single recorded value. This is now specified in the text as follows (new line 192):
“The Wattbike measures chain tension with a load cell at 100 Hz and crank angular velocity twice per revolution yielding two power values per revolution [51,52]. For the present study, only the highest power value recorded during the six-second cycling sprint test (6sCS) was analyzed.”
167why? Normalization to bodyweight often has been done, this will decrease correlation coefficients.
The reason for using non-normalized data should be provided and the effects of normalization of r-values should be included.
We agree that using body-mass-normalized power instead of absolute power tends to decrease correlation coefficients, because the overall range of data is typically smaller. Because this issue affects single test values, this is mainly the case for our cross-sectional correlations. For this reason, cross-sectional correlations in the present study were already presented for both absolute and normalized power. On the other hand, whether power has been normalized or not has a much smaller effect on the data range of percent change scores, and thus on the correlation coefficients. With this reasoning and in order to keep data presentation concise, we did the longitudinal correlations on change scores from absolute power only. To make our reasoning in this regard more clear, I have expressed it explicitly in the first paragraph of section 2.3.
In relation to this: What is the effect of differences in body composition: including male and female athletes also increases between-subjects variability (and thereby increase correlation coefficients)
Cross-sectional correlations for males and females separately have been added to the manuscript. According to the same logic as above (gender not being expected to substantially affect percent changes) and in order to maximize n, we chose not to separate genders for the longitudinal correlations. This too is now declared explicitly in the first paragraph of section 2.3. Differences in body composition which you are concerned about would affect the spread of normalized power data (and thus correlation coefficients), but not the spread of absolute power data. Therefore, you are right: it is important to present cross-sectional correlations for normalized and absolute data, which we have done in the original manuscript.
169 …compared using a one-way ANOVA and paired t-tests. I would have thought that a repeated-measures ANOVA with two within factors (time; 6 levels) and test-type (3 levels) with Bonferroni adjusted postdoc test would be the appropriate test.
This part of the study was purely descriptive and intended to show differences between tests, not between time points. Therefore, the one-way ANOVA was the correct test. To avoid the confusion that elicited your comment, I have added to the text (beginning of section 2.3) that the values for each test are averaged across all time points to make this more clear. As for post-hoc, it is now done with Bonferroni correction as you correctly suggest. This two is now specified in the text
181 a percent typical error of 2.5% was used (based on unpublished reliability assessments from our 181 lab), 2.5% doesn’t seem unreasonable. However, it does not seem in line with the often large change scores (Figure 3)
In the revised manuscript, we have applied slightly different typical errors (2.7% and 4.7% for SJ and LSJ, respectively), this time citing a dissertation from our lab which obtained these values. As for the occasional large changes, the following has been added to the discussion (new line 367):
“Although observed changes were mostly within around ±9%, it might seem remarkable to some readers that professional athletes occasionally experienced changes of almost ±20% within a few months. However, we do not find such changes particularly surprising considering that the present subjects’ (skiers’) explosive strength is generally only moderate (compared to sprint specialists, for example). At the same time, the present subjects (and skiers in general) possess very good maximal strength, which affords their explosive strength a rather large degree of trainability during the off-season conditioning phase. On the other hand, a competitive season and a post-season break lay between time points 3 and 4, which could explain most of the larger decreases in power we observed. The occasional extreme changes, especially between consecutive time points, certainly expanded the data range, which tends to increase correlation coefficients. Accordingly, we attribute the lower correlation coefficients between change scores over one year to the narrower range of change scores (due to the fact that athletes were at the same phase of their physical preparation period at both time points).”
200 please report separate power values for men and woman. It is hard to see, but I am not particularly impressed by the Watt bike peak power values. Please provide more details about the athletes how proficient were they? Were there any junior athletes (who in general may show relatively large progress over the years etc.)
Data for females and males are now presented separately (good idea). As for the perhaps surprisingly mediocre peak cycling power values, the following has been added to the discussion (new line XX), although one should also keep in minde that the subjects were world-class skiers, not world-class sprint cyclists:
“Another explanation for the lower power values in the 6sCS (<500 W in a few cases) was the combination of low ergometer resistance and a flying start, an effect that was exacerbated in the case of athletes with particularly steep force-velocity profiles (i.e., high relative maximal strength but comparatively maximal contraction velocity). In essence, the movement velocity was seemingly too high for some athletes and peak power was limited accordingly. Indeed, it pilot measures performed after the study with some of the same athletes, cycle sprint tests with higher resistance and/or a standing start yielded 5-10% higher peak power values. This poses the question of whether the test protocol and resistance settings recommended by Wattbike should be applied to strength-trained athletes, which might merit further exploration in practical settings.”
Line 202: this should be a repeated-measures ANOVA with Bonferroni corrected pair comparisons I think (not that the outcome probably will be different)
Bonferroni correction is applied in the revised version and, no, it did not affect the results.
Figures: please increase Font sizes along the axis for readability
Font sized have been increased in figures
Figure 2 please use clearly different symbols for the data of the men and women, this makes the increase in the between-subject variability that is created by the inclusion of both sexes, visible (which is an honest thing to do) and again 400 Watt peak power for a top athlete, doesn’t seem impressive (even for a woman, but this is hard to judge without normalization)
As you suggest, there is now a figure with body-mass-normalized power. Also, males and females are now plotted with different symbols (good idea).
Please check numbering of tables and figures in the text.
These should be correct now.
Figure 3 I see changes in performance up to 20%
This is now discussed. Please see comments above.
these are high-levels athletes? 20% changes (or even 10 %) are enormous. even within a few months. Are these athletes recovering from injury or suffering from an injury or extensive periods without training (-20%!)
This is now discussed. Please see comments above. Even if there were injuries, overtraining, or detraining in play, that would not compromise the study, whose aim was merely to assess the relationship between changes (whatever their cause) for the different tests. And after all, such factors exist in practice as well and should be reflected in performance test results.
the typical errors for these jump test only is 2.5% thus these changes seem real, but I never encountered a healthy well-trained athlete who trained for several years to increase jump performance by 10-20 %. Has this, for example, to do with the fact that peak power (instead of average power) was calculated: I can imagine that variability in peak power is larger, because it may be more sensitive to smaller changes in jump-execution (compared to average power and jump height)
This is now discussed. Please see comments above.
Especially since you emphasise in the discussion (line 308) ;
It is worth mentioning here that the present subjects were highly-trained professional athletes, the likes of which experience smaller changes in performance for a given time span than untrained subjects.
It is true that the present subjects are highly-trained, however not specialized in explosive power (compared to sprinters, for example). And indeed, large change (as you mention) occurred in the study, Therefore, this sentence has been removed, so as not to confuse readers. Also, the effects of data range on correlations is discussed in the revision (see response to comment on line 181, above).
Magnitude based inferences depend on choices (typical errors confidence intervals) that one makes. The outcome of the study, that change scores for peak power during cycling and jumping can be different, and the deviation may increase with longer intervals between measurements is clear (and seems not very surprising, but still relevant to investigate). However, the outcome very much depends on if the test- circumstances over the years (between test moments) did not change. These factors (for instance the coach being present or not, a different coach and/or test pilot being present) affect performance. How valid is the assumption of a constant typical error? Could you comment upon the stability of the testing circumstance in the discussion? Are you positive that every individual really performed maximally on every occasion (also because sometimes only 1 attempt seems to have been made)
We believe test circumstances were reproducibly and that typical errors were valid throughout the data collection period. Moreover, any factors (life, training, etc.) that may have affected performance at any given time would have either 1) affected all tests, in which case it would not be relevant for our research aims, or 2) affected one test more than others, which would make our findings more relevant for practical settings, where such influencing factors are a reality. Based on this reasoning, and to avoid the discussion being too long, we have decided this point does not necessarily need to be discussed in the text.
Most importantly: please include the analysis for the comparison between SJ and LSJ.
These are now included.
Now you compare each of them with the cycling test and in the conclusion, you state that the choice of the test should be made in relation to the athletes’ strength and condition training. (which in the present athletes would be more a jump-type of training).
If you could show that the similarity between changes scores is greater between the two jump-types, compared to between each of the jumps separately compared to cycling, this would seem to strengthen your suggestion.
CMJ data are now included and relationships between SJ and LSJ and between SJ and CMJ discussed (e.g. new line 361) as a point of reference of how well changes can be expected to agree. Also, the paragraph with average change, which discussed the accumulation of confounding factors has been removed.
On the other hand, I wouldn’t be surprised if the similarity of the change scores between both jump types would not be greater at all compared to those in relation to cycling. There always is random variability in these tests that quite easily may become larger (with all kinds of things happening, injury, illness, broken relationships ) than the real changes in performance that can be attained, especially in highly trained athletes in which progress is difficult to achieve.
It did turn out that even the various jump tests did not agree especially well with one another and this relativizes the findings somewhat. This is now discuss, e.g. in new lines 401 – 404.

Reviewer 2 Report
Authors performed an original research about the relationship between cycling test and vertical jump test of anaerobic power assessments.
Sample size is not adequate, but recording techniques are well clear.
Although this manuscript has interesting merits and accuracy, some minor issues can be addressed to improve the quality and comprehensibility. In other words, this manuscript need to some modifications and supplements as follows.
Three major comments in your research as follows;
- If the subjects of this study were skiers, why did the authors measure the variables expressed in the muscles of skiers using cycling tests and vertical jumps, despite the many simulators that can be measured while skiing?
- Is this study approved by the IRB? The IRB date and approval number must be indicated in the appropriate section.
- Squat jumping power test (especially, LSJ) can injure the ankle, knee and lower back, and it is necessary to explain whether there was no injury during the test or how to secure the test.
- There are quite a few differences in the anaerobic power of women and men. Why did you need to integrate all of them and analyze the data?
- Did you investigate the normal distribution of the data prior to direct data analysis? If the actual data do not satisfy the assumption of normality, it is good to check whether the assumption of normality is satisfied before the analysis because the validity of the statistical analysis results is poor. Since there are 20 subjects in this study, the degree of normality distribution by Shapiro-Wilk test should be tested. If normality is not recognized in the normality distribution, correlation analysis by Spearman correlation method is recommended rather than Pearson correlation.
Three minor comments in your research as follows;
- To help readers understand, it was very helpful to visualize the results. However, it is not necessary to mark the SD up and down on Figure 1. Generally, SD is indicated only on the top.
Best wishes,
Author Response
General notes to the revision:
Due to filtering, data vary slightly from those in the original manuscript. Also the criterion for contradicting inferences was re-defined and slightly stricter than before (before only ‘very likely increase’ and ‘very likely decrease’ considered contradicting, whereas now, any ‘decrease,’ regardless of whether inferred to be ‘very likely’ or ‘possible,’ is considered to contradict another inference which includes ‘increase,’ also independent of the certainty qualifier). This change led to slightly different percentages of agreeing and contradicting cases than before. However, the main conclusions of the study remain the same. Also, in response to certain reviewers’ comments, CMJ, which were also measured, are included in the study.
Also, correlations for individual time points and time spans were excluded for conciseness, and because these did not contribute substantially to the study. For the same reason, the original Table 2 was removed (these data were and are still visible in Figure 3).
Point-by-Point Responses to Reviewer 2
Authors performed an original research about the relationship between cycling test and vertical jump test of anaerobic power assessments.
Sample size is not adequate, but recording techniques are well clear.
Although this manuscript has interesting merits and accuracy, some minor issues can be addressed to improve the quality and comprehensibility. In other words, this manuscript need to some modifications and supplements as follows.
Three major comments in your research as follows;
- If the subjects of this study were skiers, why did the authors measure the variables expressed in the muscles of skiers using cycling tests and vertical jumps, despite the many simulators that can be measured while skiing?
- It was mostly happenstance that determined subject cohort of the present study. It was a group of athletes with whom we regularly perform the jump tests and who were interested in exploring the Wattbike test. Your question may be justified when considering which tests should be performed by skiers in practice. However, that is a decision which should be (and has been) made by the ski coaches. In any case, we consider this issue to be irrelevant to the aims of the present study.
- Is this study approved by the IRB? The IRB date and approval number must be indicated in the appropriate section.
- Please see section 2.1:
“The study was approved by the ethical review board of the canton o Bern, Switzerland (project ID 2018-00742).”
- Squat jumping power test (especially, LSJ) can injure the ankle, knee and lower back, and it is necessary to explain whether there was no injury during the test or how to secure the test.
- As is now stated in the text (new line 149), there was a mechanism for retaining the barbell, such that subjects did not land with the load on their shoulders. We perform several hundred of these test per year and have never had an athlete be injured during one.
- There are quite a few differences in the anaerobic power of women and men. Why did you need to integrate all of them and analyze the data?
- This is a good question, also posed by other reviewers. Accordingly, we have included separate results for men and women for the descriptive statistics and cross-sectional correlations. For the longitudinal correlations, we feel it is unnecessary to separate genders, because gender is not expected to affect percent change scores or the homogeneity of the change score data sets. For this reason and to maximize n, we pooled both genders for the longitudinal correlations, but have now designated males and females with distinct symbols in the figures.
- Did you investigate the normal distribution of the data prior to direct data analysis? If the actual data do not satisfy the assumption of normality, it is good to check whether the assumption of normality is satisfied before the analysis because the validity of the statistical analysis results is poor. Since there are 20 subjects in this study, the degree of normality distribution by Shapiro-Wilk test should be tested. If normality is not recognized in the normality distribution, correlation analysis by Spearman correlation method is recommended rather than Pearson correlation.
- Thanks for noticing this shortcoming. In the revised version, the Shapiro-Wilk test was performed on data sets before correlations and either Pearson’s or Spearman’s statistic was selected accordingly. There were indeed some non-normally distributed data sets (which are now revealed in the tables by correlation coefficients in italic font), especially where both genders were pooled (new in the revised manuscript are the analyses of genders independently).
Three minor comments in your research as follows;
- To help readers understand, it was very helpful to visualize the results. However, it is not necessary to mark the SD up and down on Figure 1. Generally, SD is indicated only on the top.
- This change has been made. Thanks for your constructive comments.

Reviewer 3 Report
Lines 28-30. Please rewrite to this or a similar format for ease in reading. “There are various sports, such as specific track and field events, cycling disciplines, gymnastics, combat sports, and others that require explosive actions like jumping, accelerating, changing direction, or launching an object or opponent contribute crucially to performance [1-6].”
Lines 33-34. Confusing sentence; “This ability can be referred to as explosive strength, or, since the coupling of force and velocity implied here is well represented by mechanical power, muscular power [12].” These are the definitions for “explosive strength”: Explosive-strength – the ability of a given muscle or group of muscles to generate muscular force at high velocities. (Siff, M. Supertraining.) and Explosive strength – the ability to produce high peak rates of force development and is related to the ability to accelerate objects, including body mass. (Stone, M, Stone, M, and Sands, W. Principles and practice of resistance training. Human Kinetics, Champaign, IL. 2007.)
Lines 42-43. Vertical jumping only describes the direction of the jump. This article describes the differences in jump styles suggest reading for clarification, Bobbert M, Gerritsen KGM, Littjens MCA, and Van Soest AJ. Why is countermovement jump height greater than squat jump height? Med Sci Sports Exerc. 28: 1402–1412, 1996. Are the author(s) discussing countermovement vertical jumps or squat jumps which should be addressed here in lieu of later in the article?
Line 97. Remove e.g. from the citations, let them stand on their own.
Lines 135-136. The two sentences can be merged for flow.
Line 138. Was the barbell a 20 kg (Eleiko) and why was the bar load used instead of % of body mass or % of 1RM? Moreover, what was the lower-body strength level of the athletes? Were they able to back squat 1.5-2.0 x body mass?
Lines 157-162. The subjects performed the 6 s cycling power test from in-motion start versus a static start which should be mentioned as this may influence power output. There are additional studies by Martin and Elmer related to the cycling power assessment.
Line 186. Word should be “decrease” I believe.
Lines 263-266. Please rewrite this confusing sentence, “For change scores based on time points separated by one year, the most likely change in 6sCS agreed with that in SJ and LSJ change scores in 59.4% and and 42.9% of cases; in 50% and 25% of cases, respectively, the probability percentage differed by less than 20.”
Line 274 & 285. At the beginning of the manuscript it states (Line 109 – “…elite strength-trained athletes”) and now they are power-trained, please correct for consistency of the study.
Also, what was criteria for strength? Squat 2x BW based on work by MH Stone?
Author Response
General notes to the revision:
Due to filtering, data vary slightly from those in the original manuscript. Also the criterion for contradicting inferences was re-defined and slightly stricter than before (before only ‘very likely increase’ and ‘very likely decrease’ considered contradicting, whereas now, any ‘decrease,’ regardless of whether inferred to be ‘very likely’ or ‘possible,’ is considered to contradict another inference which includes ‘increase,’ also independent of the certainty qualifier). This change led to slightly different percentages of agreeing and contradicting cases than before. However, the main conclusions of the study remain the same. Also, in response to certain reviewers’ comments, CMJ, which were also measured, are included in the study.
Also, correlations for individual time points and time spans were excluded for conciseness, and because these did not contribute substantially to the study. For the same reason, the original Table 2 was removed (these data were and are still visible in Figure 3).
Point-by-Point Responses to Reviewer 3
Lines 28-30. Please rewrite to this or a similar format for ease in reading. “There are various sports, such as specific track and field events, cycling disciplines, gymnastics, combat sports, and others that require explosive actions like jumping, accelerating, changing direction, or launching an object or opponent contribute crucially to performance [1-6].”
The sentence has been revised as follows (new line 29):
“In various sports—certain track and field events and cycling disciplines, gymnastics, combat sport, and most game and snow sports—explosive actions like jumping, accelerating, changing direction, or launching an object or opponent contribute crucially to performance…”
Lines 33-34. Confusing sentence; “This ability can be referred to as explosive strength, or, since the coupling of force and velocity implied here is well represented by mechanical power, muscular power [12].” These are the definitions for “explosive strength”: Explosive-strength – the ability of a given muscle or group of muscles to generate muscular force at high velocities. (Siff, M. Supertraining.) and Explosive strength – the ability to produce high peak rates of force development and is related to the ability to accelerate objects, including body mass. (Stone, M, Stone, M, and Sands, W. Principles and practice of resistance training. Human Kinetics, Champaign, IL. 2007.)
The sentence and the preceding one have been revised as follows (new line 31):
“Actions such as these depend heavily on the ability to generate muscle and external force at high velocities and within timespans ranging from several milliseconds to a few seconds [3,7-11]. This ability can be referred to as explosive strength [12]. Since the coupling of force and velocity implied here is well represented by mechanical power, and because the energy for muscle work during these short-duration actions is not generated aerobically, the ability can also be referred to as muscular power [13] or anaerobic power [14].”
Lines 42-43. Vertical jumping only describes the direction of the jump. This article describes the differences in jump styles suggest reading for clarification, Bobbert M, Gerritsen KGM, Littjens MCA, and Van Soest AJ. Why is countermovement jump height greater than squat jump height? Med Sci Sports Exerc. 28: 1402–1412, 1996. Are the author(s) discussing countermovement vertical jumps or squat jumps which should be addressed here in lieu of later in the article?
You are correct that “vertical jump” is a very broad term, including many possible variations. For precisely this reason, the sentence, which refers to a wealth of past research, used the term vertical jump tests, without specifying which ones. To be more clear about this intention and in response to your comment, the text now reads (new line 43): “…various forms of vertical jumping…”
Line 97. Remove e.g. from the citations, let them stand on their own.
“e.g.” has been removed from citations.
Lines 135-136. The two sentences can be merged for flow.
Text now reads (new line 145): “…single vertical jumps, including countermovement…”
Line 138. Was the barbell a 20 kg (Eleiko) and why was the bar load used instead of % of body mass or % of 1RM? Moreover, what was the lower-body strength level of the athletes? Were they able to back squat 1.5-2.0 x body mass?
The (10-kg) barbell was loaded with weight plates according to the required load (20, 40, 60, 80, or 100% of body weight). For better clarity, the text now reads (new line 148): “…10-kg barbell loaded with weight plates…”
For classification of subjects as “strength-trained”, section 2.1 (new line 128) now includes the following:
“Estimates of subjects’ one-repetition maximum for back half squats (knee angle ~100°) based on isometric squatting against a force plate and a validated conversion factor of 0.7 [48] were 2.0 – 2.7 × body mass for females and 2.2 – 3.4 × body mass for males. Thus, subjects were considered to be strength-trained. “
These estimates include values assessed shortly after athletes’ post-season training break.
Lines 157-162. The subjects performed the 6 s cycling power test from in-motion start versus a static start which should be mentioned as this may influence power output. There are additional studies by Martin and Elmer related to the cycling power assessment.
To be more clear that it was a flying start, the text now reads (new line 180):
“Following the last vertical jump, athletes proceeded to the six-second cycling sprint (6sCS) test. This was performed with a flying start on a Wattbike…”
We are aware that a flying vs. a standing start affects peak power. In the revised manuscript this is discussed further (new line 337), also in response to another reviewer’s comment.
Line 186. Word should be “decrease” I believe.
You are right. The correction was made, but in the end, this sentence was re-written altogether in the revised manuscript. Thanks for the observation.
Lines 263-266. Please rewrite this confusing sentence, “For change scores based on time points separated by one year, the most likely change in 6sCS agreed with that in SJ and LSJ change scores in 59.4% and and 42.9% of cases; in 50% and 25% of cases, respectively, the probability percentage differed by less than 20.”
Actually, this paragraph is not necessary, since exactly the same data are reported in Figure 5. Therefore, the complicated text has been removed.
Line 274 & 285. At the beginning of the manuscript it states (Line 109 – “…elite strength-trained athletes”) and now they are power-trained, please correct for consistency of the study.
Thanks for noticing this. All instances of “power-trained” have now been replaced with “strength-trained.”
Also, what was criteria for strength? Squat 2x BW based on work by MH Stone?
Please see response to comment on line 138.

Round 2
Reviewer 1 Report
Thank you for the adequate responses. I noticed one typo error: line 172 'are were'
Reviewer 2 Report
Dear authors
You had a hard time answering all the questions properly.
I wish your paper had good results.
Reviewer 3 Report
The manuscript reads much better and is good for publication